# The Spectrum of Light Emitted by LED Using a CMOS Sensor-Based Digital Camera and Its Application

**DOI:** 10.3390/s22176418

**Published:** 2022-08-25

**Authors:** Hyeon-Woo Park, Ji-Won Choi, Ji-Young Choi, Kyung-Kwang Joo, Na-Ri Kim

**Affiliations:** 1Center for Precision Neutrino Research, Department of Physics, Chonnam National University, Yongbong-ro 77, Puk-gu, Gwangju 61186, Korea; 2Department of Fire Safety, Seoyeong University, Seogang-ro 1, Puk-gu, Gwangju 61268, Korea

**Keywords:** color space, hue and wavelength curve, complementary metal oxide semiconductor sensor, Bayer color filter array, digital camera

## Abstract

We introduced a digital photo image analysis in color space to estimate the spectrum of fluor components dissolved in a liquid scintillator sample through the hue and wavelength relationship. Complementary metal oxide semiconductor (CMOS) image sensors with Bayer color filter array (CFA) technology in the digital camera were used to reconstruct and decode color images. Hue and wavelength are closely related. To date, no literature has reported the hue and wavelength relationship measurements, especially for blue or close to the UV region. The non-linear hue and wavelength relationship in the blue region was investigated using a light emitting diode source. We focused on this wavelength region, because the maximum quantum efficiency of the bi-alkali photomultiplier tube (PMT) is around 430 nm. It is necessary to have a good understanding of this wavelength region in PMT-based experiments. The CMOS Bayer CFA approach was sufficient to estimate the fluor emission spectrum in the liquid scintillator sample without using an expensive spectrophotometer.

## 1. Introduction

The liquid scintillator (LS) is a mixture of an organic base solvent and fluor [1,2,3]. The organic solvent carries out the bulk of the energy absorption. Dissolved in the solvent, molecules of fluor convert the absorbed energy into light. The emitted light is generally read by a photomultiplier tube (PMT) and the maximum quantum efficiency (QE) of the bi-alkali PMT is around 420 nm. A fluor can be added to match the maximum QE of the PMT. In our study, 2,5-diphenyloxazole (C_15_H_11_NO, PPO) was used as a primary fluor, and 1,4-bis(5-phenyl-2-oxazolyl)benzene (C_24_H_16_N_2_O_2_, POPOP), or 1,4-bis(2-methylstyryl)benzene (C_24_H_22_, bis-MSB) was tested as a secondary fluor. Traditionally, the absorption or emission wavelength of these fluors can be measured with the help of a UV/Vis spectrometer or fluorescence spectrophotometer. These optical devices involved are very expensive, and, when measuring physical variables, require careful handling.

In this study, we sought to obtain the emission spectrum emitted from LS. For this purpose, instead of using a conventional spectrometer, color image processing was performed using CMOS sensors technology. After irradiating UV light on LS sample, a digital photo image in color space was taken by a CMOS Bayer CFA-based digital camera.

Meanwhile, it is necessary to understand the several color spaces needed for our study. In 1931, the International Commission on Illumination (CIE) defined RGB color space [4,5,6]. The RGB space is a three-dimensional color space whose components are red (R), green (G), and blue (B) light. The trichromatic values (R, G, B) combine together to reproduce a broad band of colors in the visible region. The CIE color space is used today not only as a standard to define colors, but also as a reference for other color spaces. By combining (R, G, B) values, the CIE model can reproduce almost any color that the human eye can perceive. Alternatively, hue (H), saturation (S), value (V) is another representation of the RGB space. According to the HSV model, color is not defined as a simple combination of the addition or subtraction of primary (R, G, B) colors, rather it is defined by a non-linear mathematical transformation [7]. Physically, H is related to wavelength. S describes how much gray is contained in a specific color and V represents the brightness. In our study, S and V values were used to reject backgrounds. If the RGB values of each pixel are known, the HSV value can be obtained from the RGB information. Then, the H value information can be converted to the corresponding wavelength. Our aim is to estimate the emission spectrum of fluors through the hue and wavelength (H–W) mapping relation from color images using a digital camera based on CMOS sensors technology.

## 2. Motivation

There are two major motivations to this paper. Firstly, according to the previous results [8,9,10,11], the H–W relation was assumed to be approximately linear, but we want to find a more accurate relationship down to the blue wavelength region. Once RGB values are known, RGB values can be converted to HSV values, and the wavelength can be obtained with the following linear approximation equation [8,9,10,11];
(1)$$\mathrm{Wavelength}\;≅\;A-\left(\frac{C}{B}\right)\times \left(H\right),$$
where, A is the endpoint of visible light, spanning 620~700 nm, B is the maximum value of hue, which is 270, C is a coefficient that converts hue to wavelength, and is in the range 170 and 340 nm, and H represents a limited hue value in the range 0~270 [12,13,14,15,16]. Since neighboring colors are not clearly distinguishable, it was proposed that, when converting to wavelength, approximations be used.

It was measured that the H–W relationship is not linear in the red and green wavelength regions [8,9,10,11]. Furthermore, since the blue color wavelength band that is mainly used in high-energy experiments has not been measured, it is necessary to measure the wavelength region below 450 nm. Through the H–W mapping relation of the emission spectrum from LS, we want to estimate what kind of fluors are dissolved in the sample. The wavelength was obtained through image analysis of the photo obtained using a digital camera based on CMOS sensors. Even without using an optical spectrophotometer, information about the emission wavelength of fluors dissolved in LS can be estimated by the H–W relationship. A mobile phone employing a CMOS sensor can even be used as an alternative to the optical spectrophotometer.

Secondly, the emission spectra of PPO, POPOP, and bis-MSB lie in the blue region [17]. To date, there has been no study in the literature of the emission spectra, based on H–W relationship especially for the blue region, or the region close to UV. It was mainly measured in the green or red regions [8,9,10,11]. As already mentioned, most of the PMTs used in experimental high energy physics or neutrino experiments have an optimized QE near ~430 nm. An expensive UV/Vis spectrometer or fluorescence spectrophotometer are commonly used to obtain information on the absorption and emission spectra of fluors. However, there is no need to distinguish the wavelength of light entering the PMT down to a few nanometers, though it is necessary to know its information in this wavelength region at a reasonable level [18]. The region of main peaks where the fluors emitted lights coincided with the PMT’s maximum QE band. Therefore, we considered a method to easily identify the fluor emission spectrums by decoding digital photo images through the H–W relationship.

## 3. Experimental Setup

Figure 1 shows the experimental setup for generating light and taking pictures with a digital camera (EOS 450D, manufacturer: Canon, Tokyo, Japan). The experiment was performed using a 4-pin light emitting diode (LED) including a minus cathode pin. For higher light intensity, a module in which several LEDs are bundled into one is used. This LED disc (Light Disc with 7 SMD RGB LED, manufacturer: DFROBOT, Shanghai, China) was connected to a single-board Arduino module [19]. Arduino (Arduino Uno (R3), manufacturer: Arduino, Italy) is a device equipped with microcontroller kits that provides open-source hardware and software. RGB values can be changed in units of 1 nm. Wavelengths from 380 to 780 nm were generated. Each RGB color light was emitted from the LED board, and each light must be thoroughly mixed. A black foam board with good reflectivity was used as an L-shaped wall. To prevent external background light from entering the camera lens, the digital camera was positioned at 90 degrees to the axis of the LED source, as shown in Figure 1b. For the background rejection and image calibration, experiments related to distance and refraction were performed in a dark room that excluded any ambient or stray light. After the light was reflected off the wall of the black form board, only the well-mixed desired light reaches the camera. Figure 1c shows RGB color in units of wavelength of 1 nm. By assigning each RGB value to color pins of the LED disc, the desired wavelength can be generated. For example, if (0, 182, 255) is assigned to the R, G, B value in the LED, respectively, light with a wavelength of 473 nm is generated. For calibration, both a colorimeter and three laser modules with wavelengths of 375, 405, and 440 nm were used. With this method, the wavelength corresponding to each color can be generated. When taken with a digital camera, color images appear as shown in Figure 1d. Photographs were taken with wavelengths ranging from 380 to 650 nm at intervals of 10 nm. The response of the camera used was not sensitive to wavelengths that exceeded 650 nm.

Organic base solvents have been widely used in LS, but recently, research on the water-based liquid scintillator (WbLS) is in progress for the future high energy experiments [20,21,22]. For this study, we used a liquid scintillator using water as the main base solvent. In the case of WbLS, fluors do not dissolve directly in water. Therefore, with the help of the appropriate surfactant, fluor can be mixed with water. A fluor was directly dissolved into a surfactant to make a solution. Then, this solution was slowly diluted with pure water to reach the desired concentration, based on the hydrophilic–lipophilic balance (HLB) index [23]. For a surfactant, ethoxylated octylphenol (C_32_H_56_O_10_, Triton X100) was used. The ratio of water to surfactant was (7:3). For detailed surfactants for the WbLS, they were well described in Ref. [24]. The amount of optimized primary fluor PPO was ~3 g/L, while the secondary WLS was ~30 mg/L, and the same fluor concentration used in the reactor neutrino experiments was maintained [2,3,20,21,22]. Figure 2 shows the experimental setting. A cylindrical quartz container filled with WbLS was placed on top of the rotating disk. WbLS samples were illuminated by a UV lamp from the top of the container at ~250, 310, and 360 nm. We tried to prevent any UV lamp light from entering the camera.

## 4. CMOS CFA-Based Digital Image Sensor

### 4.1. Bayer Color Filter Array (CFA) Image Sensor Technology

For the image analysis, a commercially available digital camera equipped with a CMOS image sensor was used. In our study, we used a 12 mega-pixel mobile phone (Samsung Galaxy S-series, SM-G973N, manufacturer: Samsung, Seoul, Korea) or a digital single lens reflex (DSLR, Canon EOS 450D, manufacturer: Canon, Tokyo, Japan) camera. Each pixel of most commercial CMOS image sensors is covered by a CFA. Each pixel receives only a specific range of wavelengths according to the spectral transmittance of the filter. The CFA configuration in the CMOS is a Bayer CFA consisting of R, G, and B filters and covers a broad band of color space. In CFA, each pixel captures just one color among R, G, or B. The other missing two color values are estimated through an demosaicing interpolation process [25,26]. There are many proposed demosaicing algorithms [27,28,29,30,31]. Among them, Bayer CFA is widely used and shown in Figure 3a. Basically, the missing unknown data for each channel are estimated based on neighboring pixel information. Nevertheless, due to the lack of information during the demosaicing process, the original color decomposed into three color filters cannot be fully restored.

Figure 3b is a typical image processing pipeline of digital camera [32]. An image pipeline is the set of components commonly used for the digital image process consisting of several distinct processing blocks. It plays a key role in digital camera systems by generating a digital color image. When we take a picture, it is initially saved as raw image data [33,34]. They are minimally processed data from the image sensor. The raw data files created by a digital camera contain a CFA image recorded by the photo-sensor of the camera. Each pixel of raw data is the amount of light captured by the corresponding camera photo-sensor. A further process of generating jpeg digital images with raw data is performed.

### 4.2. Hue–Wavelength Response Using CMOS

Figure 4 shows the mapping between wavelength and hue for the Canon 300D camera employing CMOS sensor CFA technology for the wavelength from 500 to 650 nm [9,10]. This graph shows several features. There are plateau regions with wavelength in the range of 530~560 nm and over 600 nm. At the near end of wavelengths 530~560 nm, it looks like a step-like H–W response. These features are a direct result of the CFA color filters used in the CMOS sensor. The CFA arrangement made these patterns in the H–W curve. At 120° hue (wavelength ~560 nm), only the green component exists. Because neither the blue nor red filters transmit significantly in this region, the plateau naturally occurs. The sharp drop of hue to 0° over the wavelength of 620 nm is also due to the properties of CFA color filters. Since the nature of the H–W response of CFA color filters does not allow wavelengths of a few nanometers scale to be distinguished, it should be used with caution in certain wavelength ranges. Furthermore, over 650 nm, it was not possible to obtain the H–W relation. The difference in conversion from H to W was so small that it was very difficult to distinguish neighboring values. To measure this region, we need to use a camera with a sensor sensitive to the red or IR side. Overall, Figure 4 shows that the H–W relationship was highly non-linear.

### 4.3. LED Image Analysis and H–W Response of CMOS Image Sensor

We have used a Canon EOS D series camera with CMOS image sensor for our studies. The wavelength range that can be expressed with the LED we used is from 380 to 650 nm, as shown in Figure 1c. Photographs were taken in increments of 10 nm with wavelengths ranging from 380 to 650 nm, as shown in Figure 1d. Figure 5a,c,e represents digital color images at wavelengths of 380, 590, and 630 nm, respectively. Figure 5b,d,f shows their hue distributions obtained from the color information. A dark color mixed with black appeared on the edge of the third box, so this region was rejected as a background. The image is divided into rectangular boxes to use V cut. The only V values in the first box corresponded to about 0.6 or more, and were used for the final analysis. The S value is the percentage of white light, which did not affect our analysis. In the same way, it is possible to know the hue distribution of all wavelengths in the 380 to 645 nm region that can be obtained through the RGB values in the LED.

Figure 6 shows the result of H–W mapping relation using LED light source for the wavelength from 380 nm 520 nm. Especially, we focused on around the 400 nm wavelength region, since most of the fluors used in particle physics emit light in this wavelength range. The thickness of the line represents the deviation of hue value obtained by Gaussian fitting with different exposure times for this wavelength region. Below 400 nm, the error band is relatively large compared to other ranges due to the poor camera response. Using this relationship, the emission wavelength of fluors can be estimated.

### 4.4. Fluor Emission Spectrum from a Color Image of Liquid Scintillator

Based on our LED inputs, as shown in Figure 1, our measurement was expanded to the blue region down to 380 nm. Due to the intrinsic limitations of the CMOS sensor we were using, the camera was not sensitive below 380 nm [31,32,33]. Around that wavelength region, color images were not properly reconstructed. Therefore, special attention was required in this wavelength region.

As an example, using these results, we investigated the possibility of estimating the emission spectrum of fluors dissolved in the liquid scintillator samples through the H–W relation after irradiating UV light on the sample. The camera and the sample were focused and well aligned to one another. After a photo was taken, the saved image was not raw data from the real world. It was a sampled lossy compressed image data file using only a certain number of pixels in the digital world. The R, G, and B values of each pixel were stored in 256 scale. The dimension of one pixel of a digital camera is provided by each manufacturer.

WbLS samples were illuminated by an UV lamp at ~250, 310, and 360 nm. The light emitted from WbLS samples can be clearly seen, as shown in Figure 7a. The far left was a sample filled with water. Pure water (H_2_O) did not emit any light. The far right was a green dye sample. For this, 6-carboxyfluorescein (C_21_H_12_O_7_, 6-FAM) green dye fluor was used. 6-FAM is a fluorescent dye with an absorption wavelength of ~490 nm and an emission wavelength of nearly 520 nm. This sample was used to check an emission wavelength through H–W response. Figure 7b shows a light image emitted from the 6-FAM sample. To remove backgrounds, the sample was divided into three rectangular box areas. Only those pixels whose V value was greater than 60% were selected.

Figure 8a shows the distribution of blue component of each sample in the color space as a function of the pixel intensity value. Because the fluors emit light in the blue wavelength region, blue pixel values are dominant among R, G, and B values. Our WbLS contains three fluors substances that convert from UV to visible light, PPO, POPOP, and bis-MSB [17]. In addition, a green pixel intensity of 6-FAM can be clearly seen, rather than blue pixel intensity. Figure 8b shows the extracted emission spectrum of PPO, PPO+POPOP, PPO+bis-MSB according to H–W relation. The difference of emission spectrum between PPO and POPOP (bis-MSB) fluors in the blue-like color region can be clearly distinguished. Compared to the expensive commercial UV/Vis spectrophotometer scanning roughly 200 to 700 nm, the current method has the disadvantage that it cannot measure the UV region below ~380 nm and up to 1 nm increment level. However, this method has sufficient potential to estimate the emission spectrum in the visible region [11]. Unlike the spectrophotometer, there is no need to perform extra steps, such as extracting samples from a sealed liquid container, inserting and removing the cuvette from the spectrophotometer, and thorough cleaning after measurement.

## 5. Conclusions

We investigated the H–W relationship using a digital camera based on CMOS sensor technology. To date, in the red or green color bands, it was measured relatively well, but there were no measurement results in the blue color region. Most of the signals generated in experimental particle physics or neutrino physics are read by PMTs, and the maximum QE of bi-alkali PMTs lies in the blue region. This is the reason why we are interested in the blue region of the wavelength and pay attention to the corresponding H–W relation. There are several types of image sensors. Among them, in our study, a camera employing CMOS sensors technology was used. Further follow-up studies of a Foveon-based camera will be performed in the future, because it provides a three-dimensional color filter configuration.

If the RGB value of each pixel is known, the HSV value can be obtained from the RGB information. Then, H value information can be converted to a wavelength. An H–W value was measured around the 400 nm wavelength using LED. A non-linear relation in H–W was investigated for the wavelength around 400 nm.

In addition, photo images of UV lights onto the LS sample were taken by a CMOS CFA-based digital camera. We considered a method to easily identify the fluor contents by analyzing the emission spectrum with an adequate precision level. The emission spectrum difference of PPO, POPOP, bis-MSB, and 6-FAM samples could clearly be seen. This simple method was sufficient to identify the fluor contents in the LS through the demosaicing process in the Bayer CFA approach. In summary, we hope that our image analysis will be used in future particle detector technology or other related fields.

## Figures and Tables

**Figure 1 sensors-22-06418-f001:**
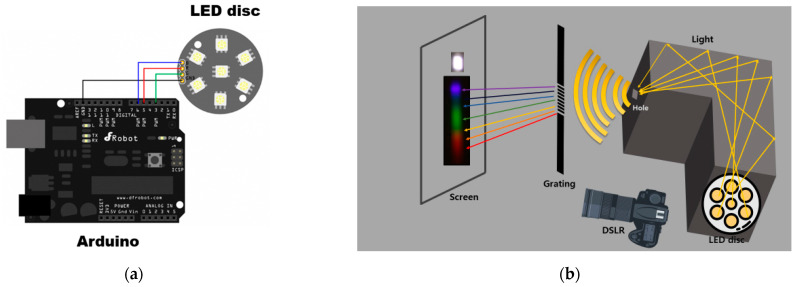
(**a**) Experimental setting for generating lights using LED disc. LED disc was connected to a single-board Arduino. (**b**) Schematic for taking digital image photo. Black foam was used as an L-shaped wall. The light from the hole passed through the grating or was directly imaged on the screen depending on the purpose of the experiment. (**c**) RGB intensity as a function of wavelength. (**d**) RGB color and corresponding wavelength. Light from blue to red was generated. Photographs were taken in increments of 10 nm with wavelengths ranging 380 to 645 nm.

**Figure 2 sensors-22-06418-f002:**
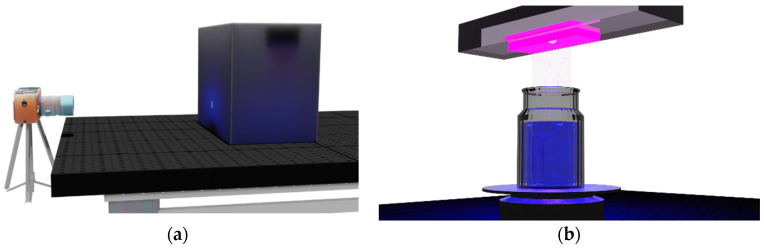
(**a**) Illustration of the experimental setup for taking digital images. A digital camera was remotely controlled. The WbLS sample was placed in a black box. There was a hole in the front of the box, and the light coming out through hole was photographed. (**b**) Schematic diagram showing the inside of the black box. A cylindrical quartz container was placed on top of the rotating disk, which can rotate at constant speed. UV light illuminates the sample from the top of the container to the bottom.

**Figure 3 sensors-22-06418-f003:**
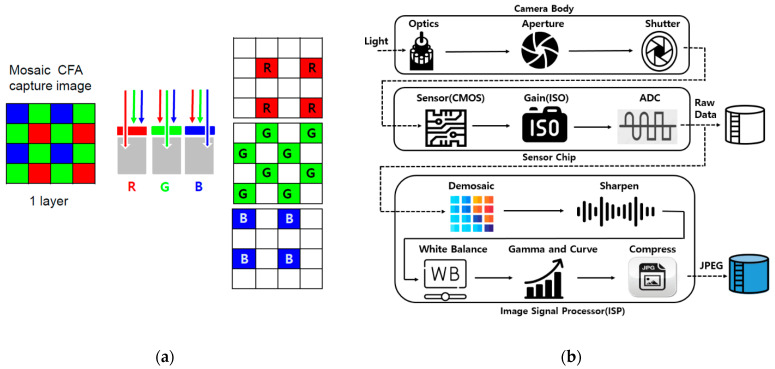
(**a**) Conventional mosaic capture CMOS sensor CFA technology. Color filters are applied to a single layer of pixel sensors, and make a mosaic pattern. Only one wavelength of R, G, or B passes through a pixel, and one color is recorded. (**b**) Schematic of the color image pipeline for making raw data image and jpeg image. Each stage of an image pipeline is fairly standard; although, varying in order.

**Figure 4 sensors-22-06418-f004:**
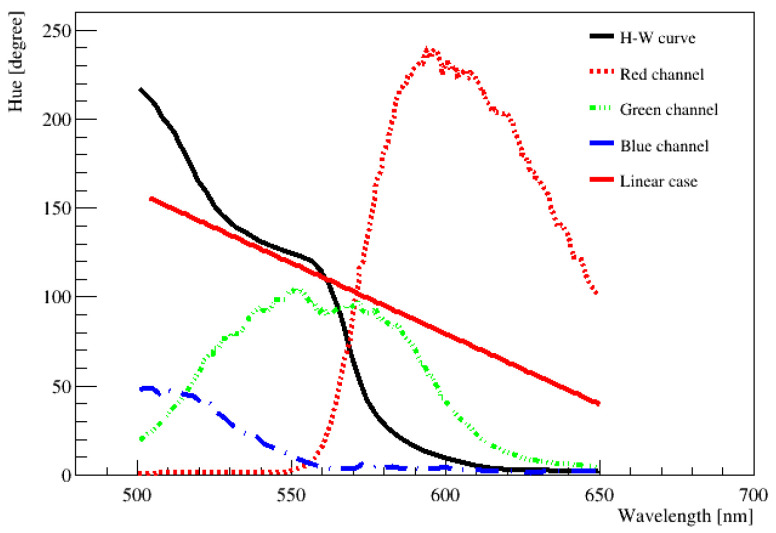
H–W response curve using CMOS CFA technology (for example, Canon 300D camera, manufacturer: Canon, Tokyo, Japan), especially in the green and red wavelength regions adapted from [9,10]. RGB curves are input values to the camera. The plateau regions with a wavelength in the range of 530~560 nm and over 600 nm occur. The red solid line is assumed to be linear [8,9,10,11]. To date, the H–W relation has not been well measured below ~500 nm, in particular, the blue color region.

**Figure 5 sensors-22-06418-f005:**
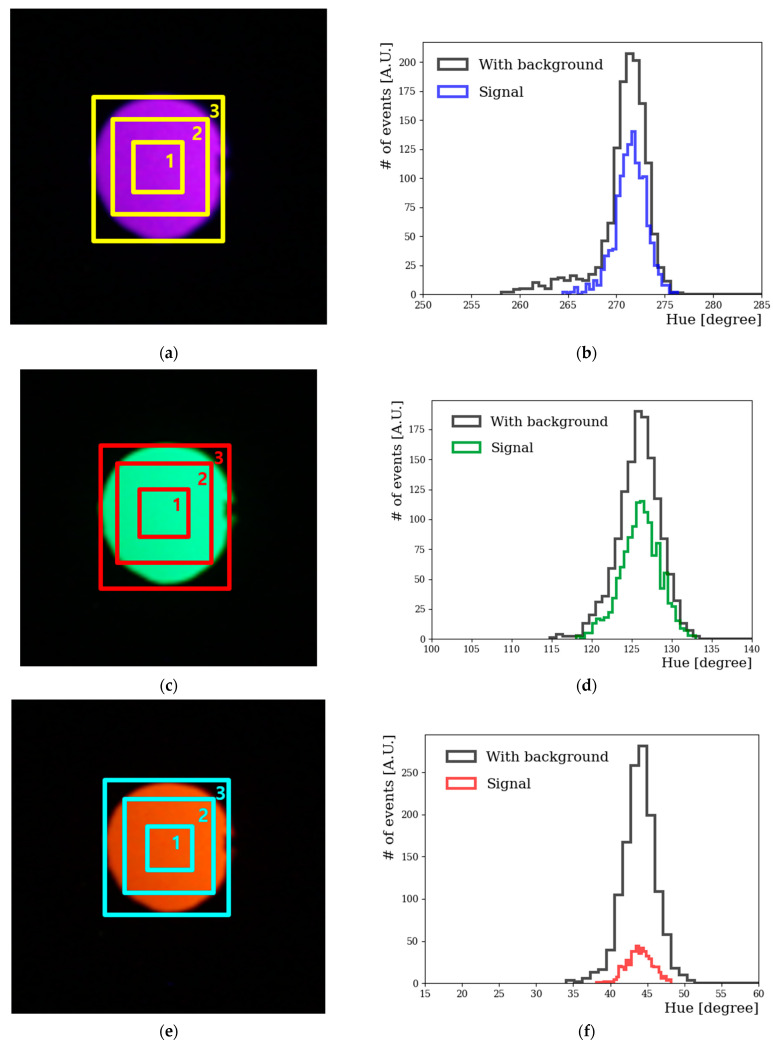
Color image (**a**,**c**,**e**) and hue distribution (**b**,**d**,**f**) with Canon EOS 450D using CFA technology at wavelengths of 380, 590, and 630 nm, respectively. The rectangular box represents the value of V, and the first box corresponds to approximately 0.6. Hue distributions are shown in (**b**,**d**,**f**) before and after the V cut was applied.

**Figure 6 sensors-22-06418-f006:**
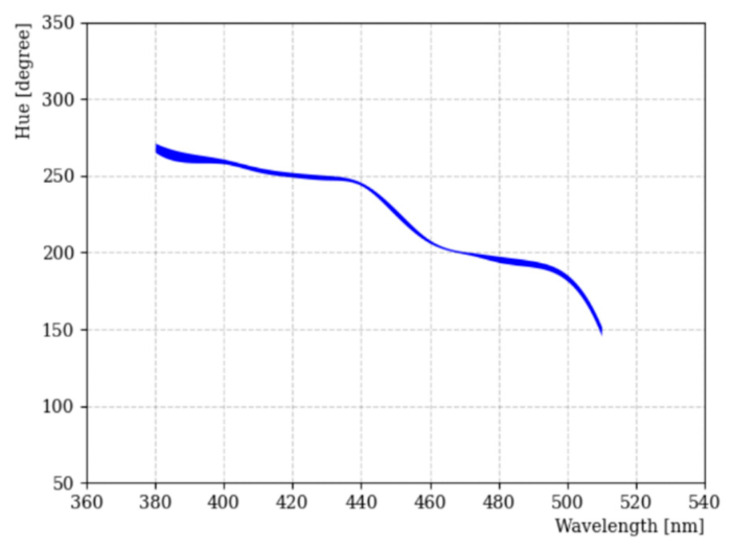
H–W curve with Canon EOS 450D using CFA technology. The thickness of the line indicates the deviation of hue across the limited dynamic range of the camera for the wavelength from 380 nm to 520 nm.

**Figure 7 sensors-22-06418-f007:**
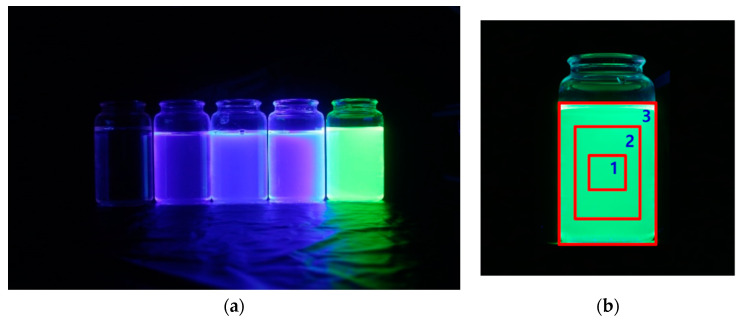
(**a**) Light is emitted from five samples illuminated by UV lamps. From left to right, pure water (H_2_O), PPO, POPOP, bis-MSB, and 6-FAM samples. Pure water did not emit any light. A cylindrical quartz container was filled with WbLS using Triton X100 surfactant. (**b**) Only those pixel regions with a V value above a certain level were selected, and their boundary lines were displayed as a rectangular box area. The first box whose V value was greater than 60% was selected for the background rejection.

**Figure 8 sensors-22-06418-f008:**
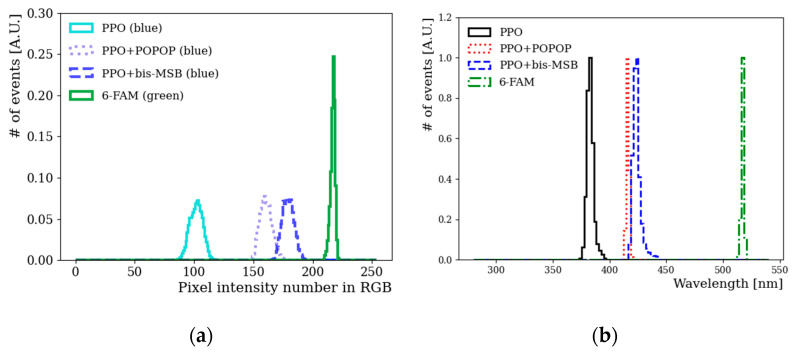
(**a**) Blue components were extracted from the photographed images of PPO, PPO+POPOP and PPO+bis-MSB. In addition, a green component of 6-FAM was shown. The spectrum of blue or green component in the RGB color space was drawn as a function of color intensity value with 256 scale. The smaller a pixel intensity value, the closer it is to black. (**b**) The extracted emission spectra of PPO, POPOP, bis-MSB, and green dye, based on H–W relation. The separation between PPO, POPOP, bis-MSB, and green dye samples can be clearly seen with a reasonable separation.

## Data Availability

Not applicable.

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
