# Peer review of "The Spectrum of Light Emitted by LED Using a CMOS Sensor-Based Digital Camera and Its Application"

_sensors, 2022, doi:10.3390/s22176418_

Round 1
Reviewer 1 Report
Review of the paper of the title “ Emission spectrum discrimination using a color image with
3 LED and a CMOS sensor-based digital camera” submitted for Sensor Journal MPDI
The paper describes the analyze of the spectrum of light emitted by LED using digital camera, so in my opinion the title should be changed to be suitable to the content of the paper.
Generally the text is understood in English, but several phrases should be written or more clearly explained.
Lines 23-24: keywords are too much and abbreviation should not be used here.
Line 55: The parameters S and V should be more explained
Line 126: If we have a light with known wavelength, we can determine a values of R,G and B ? By other words: H is a unambiguous function of R,G and B ?.
Lines 173-174: The phrases: “Since each pixel only captures one-third of the color, mosaic-based image sensors rely on complex processing to interpolate the remaining two-thirds of the colors that are not captured” This phrase is not clear and requires more explanation.
Line 202. The caption of the figure 4 is missing
Lines:303-306: The lines 303-307 is not clear, and caption of figure 7 is missing.
Line 344: similar to the figure 7, the caption of figure 8 is missing.
Line 372-373: Based on the statement at lines 372-373, the question is due to non-linear relation of H-W, do we receive many values of H for one complex values of RGB?
Author Response
Please take a look at an attached file.
Thank you very much.

Reviewer 2 Report
Dear Authors,
below my comments.
General comment:
Your overall idea to provide an approximate wavelength measurement by using a camera is pretty cool and your method should be published. The results, which you show for different WLS in the end of the paper are quite encouraging.
However, a revision of the paper is needed. The main problem I have with the paper is that you do not show your calibration curve, i.e. a reference H-W measurement with your camera. In Figure 2b you show the different wavelength light produced by the LEDs but you do not demonstrate:
a) that this light has indeed the wavelength which you desire it to have, and
b) you never show a curve with your measurement of the H to W relation from 380 nm to 645 nm. (This is in particular bad, as you stress that there is no measurement below 480 nm and that you are actually performing this measurement in the paper. But then you do not show it.)
These two points have to be addressed in the text and I think also by adding appropriate figures. In Figure 1 and Figure 5 you show some H-W graph, but these plots appear to be from literature, as your measurement should start at 380 nm and the curves in the plots all start at 480 nm.
Detailed comments:
47: What is XYZ colour space supposed to be ?
106: In Figure 2a you show an LED ring, whilst in the text you first speak about one 4-pin LED. What hardware are you actually using ? In the text (on page 4) it seems that you have several LEDs, which would also make sense. Please clarify this in the text.
120: Do you have any external calibration for the wavelength of the RGB light you produce ? How do you know which R,G,B values to chose to produce light of a given wavelength ? Is it only the data in the plot in Figure 2 c ? If yes: Where is this data coming from ? What is the uncertainty on the wavelength of the light you produce this way ?
200: (Caption Figure 4) The caption is broken. E.g. the "Figure 4" part and at least one line is missing here.
243, following: The analysis you describe here, is basically what I would like to know. (Including Figure 6.) I would like to see an extra plot which has your final Hue measurement with your error for all the wavelengths you examine.
250: What is this V cut. Please explain. Is it that you only use box 1 as there the V is larger 0.6 ? Why is the V in box 1 and box 2 different ? It seems you only have background in box 3, but only light in box 1 and box 2.
267: Can you show your comparison with Figure 5 you are talking about ?
269: You say you can confirm the existing results and extend them, but you never show in the paper that you actually do that.
277: The look-up table you are talking about here, what is that ? An H to W table ? Or an RGB to H (and then to W) table ?
Figure 8: The caption is broken and at least one line missing.
Figure 9: The caption is broken and at least one line missing.
Figure 9a: Is this pixel intensity in RGB or is this pixel intensity only in B ?
359: "There are two types of image sensors" Where are actually a lot of image sensors - CCD, CMOS, sCMOS, film,... Do you mean that there are two methods of producing RGB images with a CMOS sensor - CFA based sensors and Foveon based sensors ?
367/368: You say you measured the H-W curve down to 380 nm, but you never show it ....
Spelling etc comments:
(These are not all, but some of them I marked during the reading. I am not a native english speaker, so you should find someone else to have a look, too.)
12: ... a liquid scintillator ....
15: ... images. ....
36: ... a UV/Vis spectrometer of a fluorescence ....
80: ... dissolved in ....
81: ... employing a CMOS sensor ....
92: ... or the region ....
94: .... an optimized ....
101/102: ... by decoding digital images through ....
105: ... generating light ....
154 (Caption Figure 3): The first sentence is not ok.
159/160 (Caption Figure 3): The last sentence is redundant
215/216: The sentence: "After taking ... image." needs fixing.
225: ... only the green ....
244: ...expressed with the ....
263: Due to the ....
269: ....result from our results.
272: ... UV light on ...
357: ... by PMTs and .... bi-alkali PMTs lies ....
365/366: Then the information ....
Author Response

(The authors gave the same response as above.)

Reviewer 3 Report
1. The first major issue in this paper is the authors have used irregular format for defining the acronyms. Some in capital letter, some are not defines, some in small and some are mixed.
2. The abstract is nor refined, it contains major flaws like hue-wavelength (H-W) and Hue and wavelength. Hue and wavelength are closely related. incomplete sentence which makes no sense.
3. The keywords must be in expanded form and not more than 5 keywords as per standard rules.
4. Please discuss a detail mathematical model for the proposed approach.
5. Add a subsection of major contributions, and comparison table
6. "There are two major motivations to this paper" please confirm if the sentence is clear.
7. Why the authors have added the Figure 1 in the paper, the description of nonlinear and linear is known to everyone. straight line and curved line, if there is something new to the readers please explain otherwise google is available.
8. Figure 2 is the experimental setup or simulation setup because the Figure looks like that it is taken from simulation software. Same comment for Figure 3.
9. The conclusion must be short and based on the achievements, no need to cited the references. Secondly, the references are cited very old. So, please make sure that recent work on the proposed model is not mentioned in current literature.
10. From the literature the authors claim that no works have done on the proposed work from 2016, however, recent publications on the proposed work are avaliable on google scholar. please confirm, if the recent work is exist then update your reference from 1931 to recent.
Author Response

(The authors gave the same response as above.)

Round 2
Reviewer 2 Report
Dear Authors,
thank you very much for your detailed answer to my comments and the changes to the paper. After this review round my main concern is (still), that you do not show any measurement of Hue vs Wavelength. I can accept that you do not want to show the full curve you measured, but if you show a plot like Fig. 7b, in which you clearly plot wavelength from 360 nm to 520 nm you have to show also the corresponding calibration to back that up. Figure 7a and 7b clearly show that you have the power to discriminate wavelengths - surely well enough to resolve them with a resolution of (more or less) 10 nm. If you want to go down that route and not show any plots with wavelength as in Fig. 7b, the paper loses some of its interest, as this wavelength by hue measurement is the really cool thing about it.
Otherwise I would suggest to include the value plot for area 1, 2 ,3 in the paper, to illustrate the selection in the analysis, but this is only a recommendation and not something you must change.
Best regards
Author Response
Hi,
Please take a look at an attached file.
Thanks for your valuable comments one more time.
Best reagrds,
Joo

Reviewer 3 Report
There are still minor issues like
1. The Figures names are not consistant like Figure 4 and Fig.4
2. Complementary Metal Oxide Semiconductor (CMOS) image sensors with Bayer color filter array (CFA) These are two abbreviations which are explained in different format. please follow same format in the whole thesis
Author Response

(The authors gave the same response as above.)
